# Intercepting of Class III Malocclusion with a Novel Mechanism Built on the Orthopaedic Appliance: A Case Report

**DOI:** 10.3390/children9060784

**Published:** 2022-05-27

**Authors:** Paolo Manzo, Maria Elena De Felice, Sara Caruso, Roberto Gatto, Silvia Caruso

**Affiliations:** 1Department of Orthodontics, University of Ferrara, Via Livatino, 9, 42124 Ferrara, Italy; paolo.manzo@gmail.com; 2Department of Life, Health and Environmental Sciences, University of L’Aquila, 67100 L’Aquila, Italy; saracaruso2704@gmail.com (S.C.); roberto.gatto@univaq.it (R.G.); silvia.caruso@univaq.it (S.C.)

**Keywords:** Class III malocclusion, orthopaedic treatment, functional appliance, paediatric dentistry

## Abstract

Aim: The following case report aims to show a novel orthopaedic appliance to reduce the side effects of the orthopaedic Class III treatment through the use of two acrylic splints combined with a PowerScope device. Materials and Methods: This case report describes the treatment of a 6-year-old patient with a skeletal Class III relationship with a maxillary deficiency and a severe hyperdivergency. The patient underwent a sagittal orthopaedic treatment with a PowerScope device for 12 months. The retention period lasted 4 months. Results: The response of the craniofacial complex to the active orthopaedic treatment of the Class III malocclusion with the PowerScope™ device splints consisted of significant changes in maxillary growth and position. Both angular and linear sagittal measurements of the maxilla showed improvements during active treatment, respectively, of 0.6° and 1.2 mm (SNA from 75.8° to 76.4°; maxillary length from 38.8 mm to 40 mm). These effects allowed for a highly significant improvement in the maxillomandibular skeletal relationships. ANB improved by 1.6° and Wits appraisal by 4 mm. Using this appliance in a hyperdivergent patient, we obtained a vertical control of the mandible with a SN/Go-Gn stable value at 40° and a significant improvement of the ANS-PNS/GoGn angle from 30° to 28°. Conclusion: The Class III orthopaedic treatment with the PowerScope™ telescopic and NiTi spring device mounted on the upper and lower resin splints in a Class III correction offered good vertical control during the early orthopaedic treatment by improving the skeletal discrepancy and controlling the hyperdivergency, which is one of the most difficult factors to control in Class III malocclusions.

## 1. Introduction

Class III malocclusion is a great challenge for orthodontists, as its aetiology is multifactorial and includes genetic, epigenetic, and environmental factors.

Angle Class III malocclusions vary greatly among and within populations, ranging from 0% to 26%. The literature shows that European countries have a lower prevalence rate of 4.9% [1].

The complexity of Class III is given by the unpredictability of the long-term stability produced by the early treatment of this dentoskeletal disharmony [2]. However, proper diagnosis and early treatment can help a growth-friendly environment [3] by decreasing the complexity and duration of the treatment in permanent dentition or by making surgical procedures less invasive when orthognathic surgery is still needed in an adult patient [4]. Early treatments of Class III malocclusions focus on control and improve dentoskeletal harmony to correct any negative overjet. Overtime, orthopaedic treatments, through the use of rapid maxillary expansion and facial mask (RME/FM), have been the gold-standard, but the loss of anchorage and the clockwise rotation of the mandible lead to seeking out alternative treatments [5]. Therefore, novel methods were conceived to avoid side effects: in particular, SEC III (splints, Class III elastics, chincup) appliances with Class III elastics and chin cap or TADs (temporary anchorage devices) with Class III elastics were tested to achieve more successful results in clinical practice [6,7].

Unfortunately, although SEC III reduces the clockwise rotation of the mandible, it requires a correct rebasing of the splints to maintain a good fitting of the appliance.

To overcome this problem, the literature proposes an appliance which, thanks to a working mechanism in compression on the two splints, prevents the continuous rebasing of the device, thus guaranteeing a correct fitting of the same [8].

In view of this, the following case report aims to show an orthopaedic treatment to reduce the side effects of Class III treatments through the use of an alternative appliance made of two acrylic splints combined with a PowerScope device. This corrector and its unique components have a very comfortable design for the patient. It consists of a telescopic system with an internal Ni–Ti spring module, which reduces treatment compliance compared to other systems, and a ball joint system, which helps lateral movement for patient comfort. The PowerScope device has been introduced in orthodontics for the no-compliance correction of Class II, since other devices are valid and aesthetic but depend on the patient’s compliance [9]. In this case, the PowerScope has been mounted in an inverted position in order to promote the upper jaw advancement and lower jaw control.

## 2. Case Report

A 6-year-old female presented in our dental clinic for a consultation presenting anterior crossbite. She presented good general health and no systemic or congenital disease. On clinical extraoral examination, the patient showed a concave profile with maxillary deficiency. Panoramic, lateral headfilm and dental cast records were taken. No clinical symptoms on articular examination were detected.

The panoramic radiograph showed the agenesis of 11 and a mixed dentition period.

The cephalometric analysis showed a skeletal Class III relationship (Wits appraisal, 9.5 mm), maxillary retrognathism (SNA 75.8°), tendency to a skeletal openbite (ANS-PNS^GoGn 30°), and hyperdivergency (SN^Go-Gn 40°). Intraorally, the patient revealed a bilateral Class III, bilateral posterior crossbite, reduced overbite and overjet and retroinclination of upper and lower incisor (Figure 1 and Figure 2).

## 3. Treatment Objectives and Alternatives

An early phase I treatment was selected in order to allow for an improvement of the transverse and sagittal skeletal relationships, to improve the soft tissue profile, guide towards a correct growth bringing the maxillary forward, correct the posterior crossbite, reach a positive overjet and overbite, and control the vertical pattern. Following the orthopaedic phase, the treatment plan went on with a retainer period with the same appliance for a limited period and then waiting for a complete permanent dentition. At a later time, in the orthodontic phase, the patient will undergo a further check-up to finalize the occlusion with the alignment and levelling phase and to manage the agenesis of the 11 dental elements.

The possibility of using the skeletal anchorage [10] to manage the agenesis is included in the consent form to avoid adverse dental effects and for the lack of dental anchorage in the anterior zone.

At the second phase of the treatment, due to the skeletal Class III and the presence of maxillary retrognathism at the beginning of the orthopaedic treatment, there could be two options for the management of the anterior agenesis:−Opening the space with implant-prosthetic rehabilitation or a Maryland bridge in the 11 area and finishing the treatment in the molar and canine class I on both sides;−Closing the space in the 11 area, ending the treatment in the molar and canine class II on the right side and molar and canine class I on the left side.

These kinds of considerations should be re-evaluated at the time of the permanent dentition and discussed with the parents to choose the best option for the patient.

## 4. Treatment Progress

At the age of 6 years, the patient started her therapy. A bonded rapid palatal expander was cemented to correct transverse maxillary crossbite. The parents of the patient were instructed to activate the expansion screw by two quarter-turns two times a day until the palatal cusps of the upper molars approximated the buccal cusps of the mandibular molars (Figure 3). 

After the correction of posterior crossbite for one month, the expander appliance was stopped, and five months after retention, the expander was removed. Impressions were taken to produce the Class III PowerScope device assembled.

This device, which is usually placed on the fixed appliance for a no compliance Class II correction, was mounted on the upper and lower removable splints in an inverted mode in order to provide Class III correction.

The patient remained for a week without any appliance to restore oral health and oral hygiene.

After the first expansion phase to recover the transverse relationship, the sagittal orthopaedic phase began. Then, the PowerScope device was inserted in the upper and lower splints (Figure 4).

The appliance consists of three components: two fixed acrylic splints and a telescopic mechanism with a Ni–Ti internal spring system (PowerScope) working in compression, allowing the correct retention of the appliance and reducing its rebasing need. The appliance was prescribed to be worn for at least 16 h per day.

The feature of this appliance is the compression mechanism that releases the forces on the two splints in the opposite direction to the removal, and this allows for a good engagement, despite being removable, and avoiding continuous rebasing.

Considering the high prevalence of caries in childhood [11,12], oral hygiene was checked at each appointment, and professional oral hygiene treatment was performed to remove any bacterial plaque accumulated on the surfaces of the teeth.

The two splints covered all the tooth crowns in both the arches. At the first insertion appointment, the modules were used to deliver a force of 260 per side in a forward direction to the upper splint and in a backward direction to the lower splint (Figure 5a,b).

The device was activated at 3-month intervals of 1 mm, increasing the force released to an orthopaedic force. During the treatment, the patient was instructed in correct oral hygiene, despite the appliance covering the entire dentition; products with a low dose of chlorhexidine were recommended to promote the health of the patient. The active phase of the treatment was finalized when the correct parameters of overjet and overbite were carried out, and after this active phase (12 months), patients were asked to use the appliance only during night hours as the retention period (for 4 months).

## 5. Results

Intraoral and extraoral pictures taken 16 months from the beginning of treatment showed an improvement of overjet and overbite and a pleasant smile (Figure 6).

The reaction of the craniofacial complex to orthopaedic treatment with the Class III PowerScope protocol consisted of significant changes in maxillary growth direction and position with an increase in SNA and ANB and favourable decrease in SNB. A post-treatment lateral cephalograph revealed an angular and linear sagittal measurement of the mandible with significant improvements during active treatment, respectively, of 0.6° and 1.2 mm (Figure 7).

These effects allowed meaningful improvement to be achieved in the maxillomandibular skeletal relationships. The sagittal values of the maxillomandibular relationships varied, and cephalometric analysis showed an improvement in ANB of +1.6° and a Wits appraisal of 4 mm. Post treatment superimpositions showed great vertical control of the mandible with a stable value of SN/Go-Gn at 40° and a great improvement in the ANS-PNS/GoGn angle from 30° to 28° (Figure 8).

Furthermore, linear measurements of the mandibular sagittal growth showed an increase of 1 mm defined by Co-Gn at the end of the treatment (Table 1).

## 6. Discussion

The aim of early orthopaedic treatment is to intercept the developing malocclusion and redirect it to physiological development. Interception of Class III malocclusion can lead to soft- and hard-tissue improvements. [13,14,15,16]. The negative anterior overjet has an unfavourable effect on the growth pattern because it could cause worse skeletal problems [17,18].

The goal of our treatment was to intercept the Class III malocclusion by improving the skeletal discrepancy and controlling the hyperdivergent skeletal pattern which is one of the aggravating factors of Class III malocclusions. The treatment with the Class III correction assembled mode PowerScope splints protocol allowed significant results to be obtained both on the sagittal and vertical planes, improving the negative overjet and controlling the vertical skeletal pattern, reaching a more pleasant profile.

### 6.1. Sagittal Changes

The advancement of the upper maxilla, control in the growth and direction of the mandible, and an improvement in the intermaxillary sagittal relationship were achieved. Maxillary measurements showed significant improvements of 0.6 and 1.2 mm or degrees. These effects allowed an improvement in the maxillomandibular skeletal relationships with an ANB that improved by 1.6° and a Wits appraisal of 4 mm. The effects of RME/FM therapy for Class III malocclusion found by Westwood et al. provided similar results [19]. Chong et al. [20] did not report significant improvement in maxillary position after FM therapy in either the short or the long term. In 2017, a systematic review showed that all selected studies reported sagittal skeletal changes, suggesting that orthopaedic therapy is effective for correcting Class III malocclusions with a low or very low level of evidence. This review underlined that the sagittal control of the mandible assessed by angular measurements suffers from the influence of a clockwise rotation of the mandible, apparently increasing the amount of the sagittal effect. In our case, the patient showed an improvement by an advancement of the upper maxilla and control of the mandibular growth without worsening the divergency and profile.

### 6.2. Vertical Changes

Using the Class III PowerScope splints in this hyperdivergent patient, we had good vertical control of the mandible.

The linear measurements of the mandible (Co-Gn) remained unchanged with a reduction in the SNB angle from 79° to 77°, and moreover, excellent control of vertical growth was highlighted with a reduction in the PNS-ANS/GoGn angle from 30° to 28°. This appliance showed a slight downward inclination of the palatal plane to SN with a reduction in the SN/PNS-ANS angle from 9.2° to 12.2°.

The favourable control of the vertical skeletal relationships produced by this appliance was probably related to the upper splint that controls the extrusion of upper molars as a side effect of the Class III treatment with FM. As opposed to Class III elastics, the vertical component of the force transferred to the arches produces an intrusion on the upper canines and lower molars that helps in controlling the mandibular clockwise rotation [21].

Furthermore, the use of the splints in maxillary and mandibular dentition has an important impact on both the sagittal and vertical planes, as they create a slide plane on which the two arches and the corresponding maxillaries can move without encountering occlusal interference.

Moreover, Ko et al. demonstrated that patients with a greater risk of relapse were the individuals with excessive backward mandibular rotation and a hyperdivergent skeletal pattern [22,23]. Furthermore, this device allowed us to conduct the treatment with extreme comfort of the child and her parents without experiencing the treatment at an early stage with social consequences [24,25].

In our case, the choice of the appliance depended on the young age of the patient, soft-tissue profile, and dento-skeletal features.

## 7. Conclusions

The use of this device, as far as it concerns one patient, allowed us to control the vertical severe growth pattern, also improving sagittal parameters despite the bad prognosis for severe hyperdivergency. Further studies with a larger sample would help to support these findings.

## 8. Limits

According to the patient’s latest records, we can say that the case is not yet over, as the patient is still growing, so it is important to maintain the results obtained without losing control of the mandibular growth. Furthermore, in the phase of permanent dentition, it will be of primary importance to manage the anterior anchorage correctly to avoid reducing the overjet.

## Figures and Tables

**Figure 1 children-09-00784-f001:**
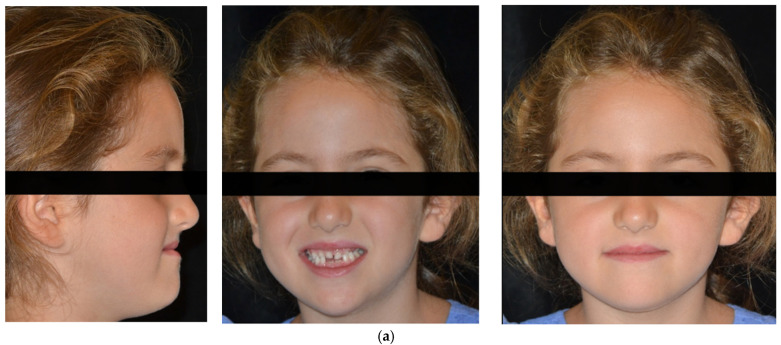
(**a**) Extraoral pre-treatment pictures: lateral and frontal views; (**b**) Intraoral pre-treatment pictures: lateral and frontal views; (**c**) Occlusal views: upper and lower arches.

**Figure 2 children-09-00784-f002:**
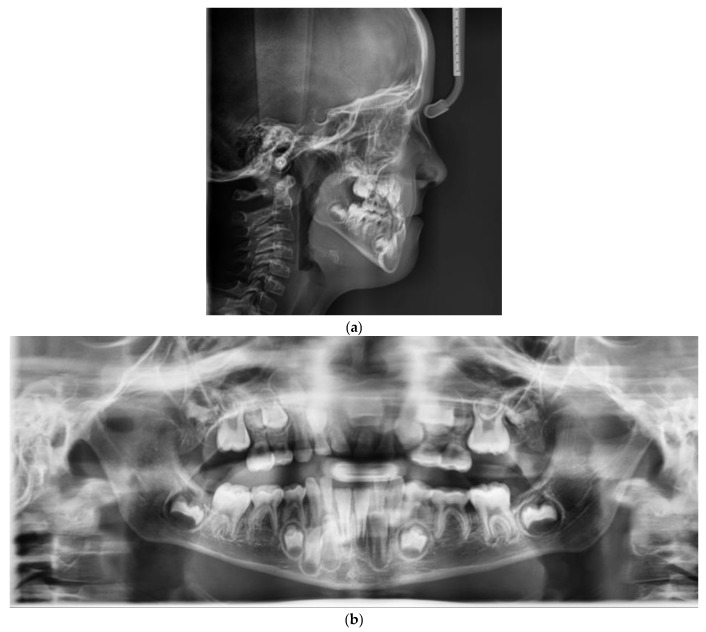
Pretreatment radiographic records: (**a**) lateral cephalogram; (**b**) orthopantomogram.

**Figure 3 children-09-00784-f003:**
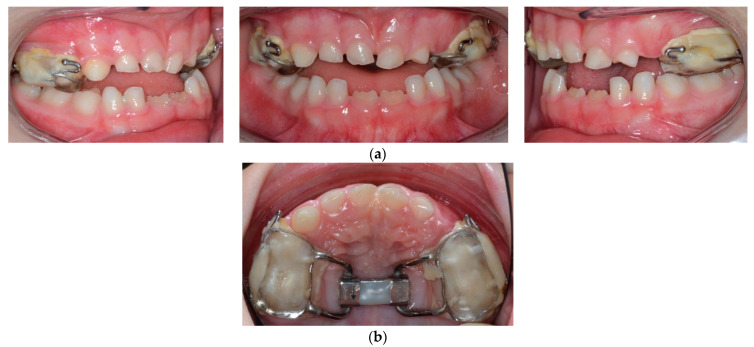
(**a**) Frontal and lateral views of a bonded rapid palatal expander; (**b**) occlusal view of the upper arch.

**Figure 4 children-09-00784-f004:**
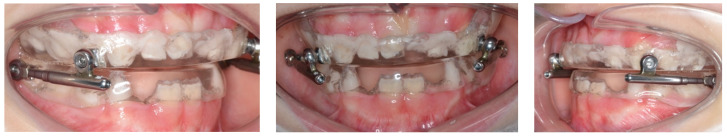
Frontal and lateral views at the beginning of the treatment with Class III PowerScope.

**Figure 5 children-09-00784-f005:**
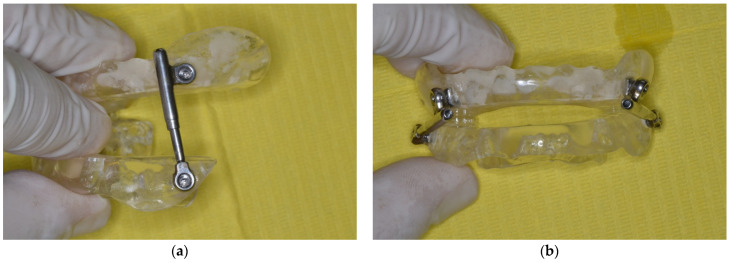
(**a**,**b**) The Class III PowerScope appliance with two fixed acrylic splints and a telescopic mechanism with a Ni–Ti internal spring system.

**Figure 6 children-09-00784-f006:**
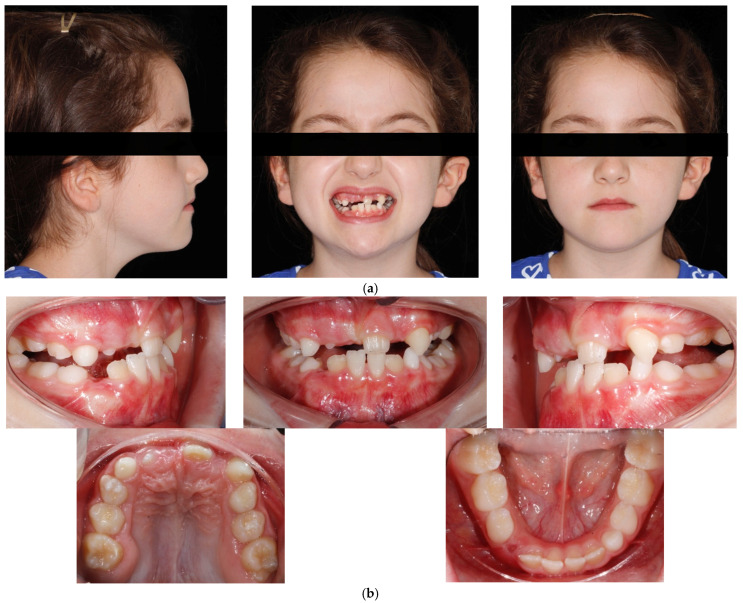
(**a**) Extraoral post-treatment pictures: lateral and frontal views; (**b**) Intraoral post-treatment pictures: lateral and frontal views; occlusal views: upper and lower arches.

**Figure 7 children-09-00784-f007:**
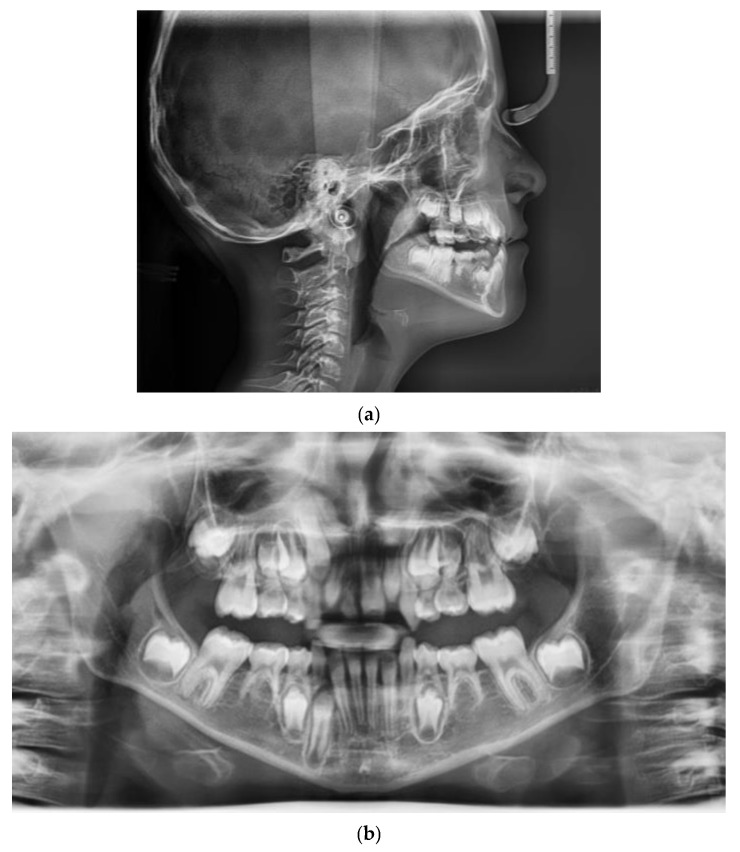
Post-treatment radiographic records: (**a**) lateral cephalogram; (**b**) orthopantomogram.

**Figure 8 children-09-00784-f008:**
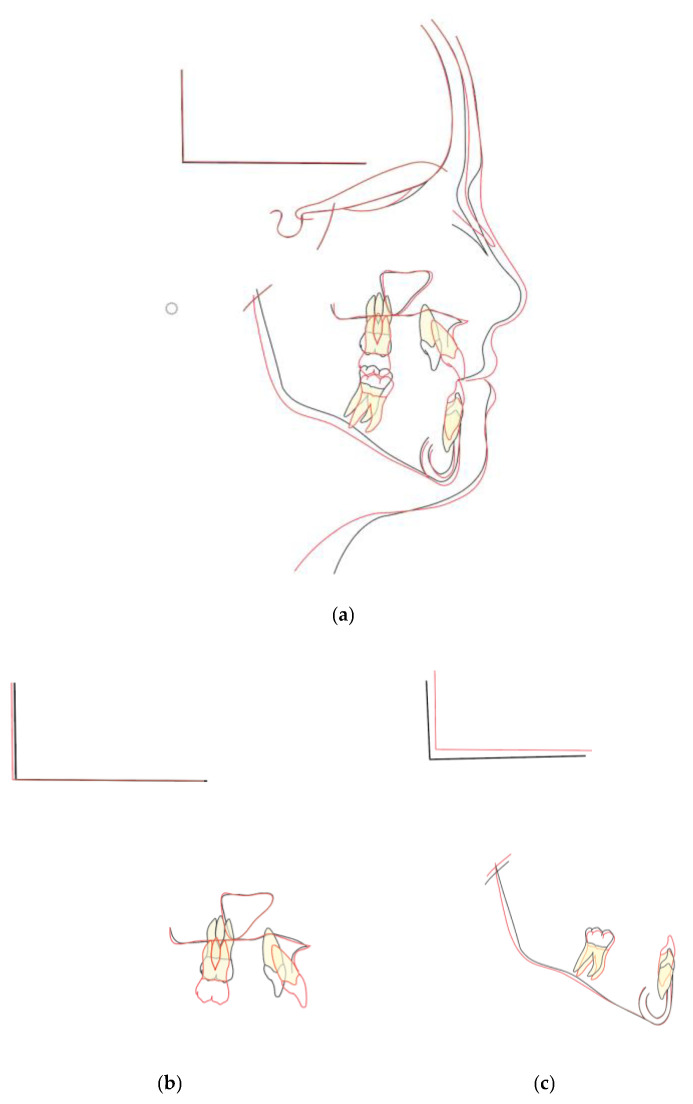
Superimposition of cephalometric tracings before treatment (black) and at the end of treatment (red): (**a**) general superimposition; (**b**) superimposition of the maxilla; (**c**) superimposition of the mandible.

**Table 1 children-09-00784-t001:** Cephalometric analysis.

	Pretreatment	Post-Treatment
Sagittal Skeletal Relations		
SNA	75.8°	76.4°
SNB	79°	77°
ANB	−3.2°	−1.6°
Wits	−8 mm	−4 mm
Vertical Skeletal Relations		
SN/ANS-PNS	9.2°	12.2°
SN/Go-Gn	40°	40°
ANS-PNS/GoGn	30°	28°
Dento-Basal Relations		
I/ANS-PNS	-	110.5°
i/GoGn	-	75°
i/APg	-	+3.3 mm
Dental Relations		
Overjet	−1 mm	+1.2 mm
Overbite	-	+0.3 mm
Interincisal Angle	-	154.8°
Linear measurements		
Maxillary length	38.8 mm	40 mm
Mandibular length	60 mm	62 mm

The absence of some measuraments in the table is due to the presence of upper deciduos incisors and unerupted lower permanent incisors at the beginning of the treatment.

## Data Availability

The authors declare that the materials are available.

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
