# Peer review of "Intercepting of Class III Malocclusion with a Novel Mechanism Built on the Orthopaedic Appliance: A Case Report"

_children, 2022, doi:10.3390/children9060784_

Round 1

Reviewer 1 Report

Comments to the Authors

  1. The manuscript could be better structured.
  2. Standard cephalometric analysis should have been used.  
  3. References need to be formatted
  4. Based on a single case report, authors should not be making such firm conclusions. It is just an observation.  
  5. Figure 8: The superimposition technique is inaccurate. Authors need to follow standard techniques such as cranial base superimposition.
  6. This article could be shortened.    

Reviewer 2 Report

This is an interesting case report on orthopedic treatment to reduce the side effects of Class III treatments through the use of two acrylic splints combined with a Power Scope device. I have some minor comments which I think the authors could address for further comprehensibility of the paper:

In the first sentence of the introduction, it would be interesting to include data on the prevalence rate of Class III malocclusion in children. In this way, the authors could better develop the epidemiological content of the paragraph.

The meaning of the acronyms must be shown at the first mention in the text. For example temporary anchorage devices (TADs), splints, Class III elastics, chincup (SEC III), on page 1 (Introduction). Please review the entire manuscript.

Change “1.1 element” to “11 tooth”. Please review the entire manuscript.

 I recommend including table 1 describing the craniofacial landmarks and cephalometric measurements evaluated in the study.

Review the legend of figure 5. Indicate the views in a and b.

In the discussion section, highlight the main findings of the case in the first sentence, relating to the purpose of the manuscript.

Further, develop ideas in the discussion. Paragraphs do not seem to have any connection (especially in "6.2 Vertical changes").

Discuss limitations of the case report design.

Indicate possible future perspectives for the use of this type of therapy, which would be interesting to investigate in future studies.

Reviewer 3 Report

The topic is of clinical interest; however, some minor changes would improve the article:

The following measurements must be added to the table.

SNB

ANB

Maxillary Length and Mandibular length must be defined.

I don’t understand whether the fixed acrylic splints were bonded or not?

If they are not bonded. They do not have enough retention.

Suppose they are bonded, how the patient uses it only during night hours as the retention period. It is not possible to use bonded appliance by herself.

In the Treatment objectives and alternatives section:

The alternative approach, such as a Face mask, Reverse chin cap, and Tongue plate must be added
